**Proposal – Lightning Talk**

**The role of Wikidata in DAMEIP Project (DAta and MEtadata for Implementing Peer Review)**

Rossana Morriello – Donatella Selva

**Abstract**

The research project DAMEIP (Data and Metadata to Implement Peer Review) is funded by the University of Florence for the years 2025–2026. It will be carried out by a research team consisting of Rossana Morriello, a tenure-track researcher (RTD-b) in the academic field HIST-04/C - Archival Science, Bibliography, and Library Science at the Department of History, Archaeology, Geography, Arts, and Performing Arts (SAGAS); Donatella Selva, a tenure-track researcher (RTD-b) in the field GSPS-06/A - Sociology of Cultural and Communicative Processes at the Department of Political and Social Sciences (DSPS), and two research fellows in their respective academic fields.
The aim of the research is to analyze the dynamics and practices of peer review in the journals of Florence University Press (FUP), selected as a significant sample of Italian academic publishing. With full respect for privacy and all related ethical aspects, the project seeks to identify and analyze patterns, procedures, quantitative measurements, and timelines of peer review, with particular attention to gender differences.
Peer review is one of the research evaluation systems that operates in two phases of the research cycle: prior to the publication of research results in a journal or other type of publication, and later in research evaluation systems implemented by national evaluation agencies, such as ANVUR. This approach is predominantly applied in the humanities and social sciences (HSS) and partially in STEM disciplines.
As part of its planned activities, the project also includes updating the essential metadata of FUP journals in Wikidata, with a focus, in this case as well, on gender representation, which is sometimes absent or ambiguous due to the common practice of using initials for authors' first names.
Wikidata is a crucial reference tool for projects and applications requiring metadata, from the simplest to the most complex, which today increasingly rely on artificial intelligence. The quality of data in Wikidata is therefore an essential element for effective knowledge organization in the digital world and in scholarly communication. Adding accurate metadata to publications also facilitates their global findability and the assignment of reviewers in internal journal processes.
The contribution of the two researchers, in the form of a lightning talk, will aim to present the ongoing project, which will officially start in January 2025 and will therefore not yet have results suitable for a full paper. Nevertheless, we believe it is important to begin disseminating it, particularly regarding the project component related to Wikidata, and the June conference "Wikidata and Research" is certainly the appropriate venue for this purpose.

**Rossana Morriello**

RTD-B tenure track Assistant professor in Archival Science, Bibliography, and Library Science (disciplinary sector HIST-04/C, formerly M-STO/08) at the Department of History, Archaeology, Geography, Art, and Performing Arts (SAGAS) at the University of Florence. She teaches Library Science, Digital libraries, Cataloguing and Metadata.
I obtained the National Scientific Qualification (ASN) as Associate Professor in April 2021.
I am the author or co-author of over 150 publications, including books, articles in national and international journals, contributions to edited volumes, and conference proceedings. I have taught at

various universities, including Sapienza University of Rome, the University of Udine, and the University of Turin.

I am a member of the scientific/editorial boards of the journals *JLIS.it* (ANVUR Class A), *Biblioteche oggi Trends* (ANVUR Class A), *Biblioteche oggi* (ANVUR scientific journal), the U.S.-based journal *Against the Grain*, and *DigitCult – Scientific Journal on Digital Cultures*. I also serve on the scientific committee of AIB (Italian Library Association) publications for monographs and contribute to the editorial team of the AIUCD (Italian Association for Digital Humanities and Digital Culture) blog.

I am a member of the Italian Society of Bibliographic and Library Sciences (SISBB), AIUCD, and the Europeana Network Association (ENA). Additionally, I am a member of the Donne 4.0 Association, whose mission is to bridge the gender gap through technology, and I participate in the working group on Goal 5 - Gender Equality for ASviS (Italian Alliance for Sustainable Development).

I am the winner of a competitive call at the University of Florence, funded by the Ministry of University, for the DAMEIP (Data and Metadata for Implementing Peer Review) project, of which I am the coordinator.

My research interests include digital and data librarianship, scholarly communication, sustainable development in cultural and library contexts, with a particular focus on issues of diversity, equity, and inclusion (DEI).

**Donatella Selva**

RTD-B tenure track Assistant professor since November 2022 in the field GSPS-06/A (formerly SPS/08) Sociology of Cultural and Communicative Processes, at the Department of Political and Social Sciences (DSPS) of the University of Florence.

Qualified as an Associate Professor on 1/06/2022.

Recipient of the Seal of Excellence from the European Commission under the Horizon Europe Marie Skłodowska-Curie Actions, call HORIZON-MSCA-2021-PF-01-01 — MSCA Postdoctoral Fellowships 2021, for the project 101065026 — EMBRACE: *"EMBRACE – Emotions Binding Race: The Emotionalisation of Racism and Anti-Racism in Digital Spaces"*.

Senior member of the Observatory on Gender, Inclusion, and Democracy, based at LUISS University.

I have taught at several universities, including LUISS Guido Carli University (teaching the course *Crisis Communication* for the English-language degree program in *International Relations*), the Pontifical Gregorian University (teaching *General Sociology* and *Contemporary Social and Political Theories*), and the University of Tuscia (teaching *General Sociology and Security*, *Foundations of Intelligence and Analysis of Political-Religious Radicalisms*, and *Strategies of Criminalisation and Processes of Radicalisation*).

Previously, I conducted research as a postdoctoral fellow at the University of Tuscia (2020–2022) and at LUISS Guido Carli University (2015–2018). At LUISS, I participated in projects of the Centre for Conflict and Participation Studies (directed by Prof. Michele Sorice), particularly contributing to research led by Prof. Emiliana De Blasio on topics such as open government, open data platforms, and participation.

In recent years, I have focused on the analysis of emotional repertoires in political communication, informational disorders, and the emotionalisation of the public sphere.