# OpenReview forum: "The role of Wikidata in DAMEIP Project (DAta and MEtadata for Implementing Peer Review)"
_wikimedia.it/Wikidata_and_Research/2025/Conference — WD&R Poster_

### Official Review · ~Franco_Bagnoli1 · 2025-01-01
**An interesting projects and a possible useful starting point for a collaborative initiative**

**Originality:** 4
**Impact:** 4
**Confidence:** 3

**Review:**

Authors want to illustrate the scopes and methodology of a starting project about metadata and peer review in the FUP journals. I think that the proposal is fully in line with the goals of the conference, that the project is innovative and can have an impact on the Wikimedia community. Moreover, in my opinion this lightning talk can originate useful discussions among the authors and the attendees, possibly originating a collaborative cooperation beyond the staff directly involved in the project.

**Compliance:**

5

**Scientific Quality:**

4

---

### Official Review · ~Iolanda_Pensa1 · 2025-01-12
**Research assessment**

**Originality:** 5
**Impact:** 4
**Confidence:** 3

**Review:**

Research assessment and peer review are topics of great interest and are at the centre of many discussions and research. Exploring the synergies with Wikidata can certainly be important. I am unsure of how much Wikidata can facilitate peer review and can host relevant data. But maybe this is a useful discussion to launch.

It would also be important to reflect on how contributions to open peer projects such as the Wikimedia projects (but also OpenStreetMap, open software, open hardware... with their open science infrastructures and practices and peer review systems) can be acknowledged in research assessment.

**Compliance:**

3

**Notes:**

I suggest that members of the scientific committee present their proposals as posters.

**Scientific Quality:**

4

---

### Decision · Program_Chairs · 2025-02-05

Accept (Poster)